# Requirement of Non-Phytate Phosphorus in 1- to 28-Day-Old Geese Based on Growth Performance, Serum Variables, and Bone Characteristics

Ning Li [1], Yuanjing Chen [1], Lei Xu [1,*], Guoqiang Su [1], Zhiyue Wang [1,2] and Haiming Yang [1]

1   College of Animal Science and Technology, Yangzhou University, Yangzhou 225009, China;
    mz120191051@stu.yzu.edu.cn (N.L.); mx120170656@yzu.edu.cn (Y.C.); mx120190671@stu.yzu.edu.cn (G.S.);
    zywang@yzu.edu.cn (Z.W.); hmyang@yzu.edu.cn (H.Y.)
2   Joint International Research Laboratory of Agriculture and Agri-Product Safety of Ministry of Education of
    China, Yangzhou University, Yangzhou 225009, China
*   Correspondence: xlei@yzu.edu.cn; Tel.: +86-514-8797-9045

**Abstract:** The standard of dietary non-phytate phosphorus (NPP) requirement is important for geese production. However, limited reports remain an obstacle to the NPP requirement of geese. We aimed to evaluate the NPP requirement in geese based on the effects of NPP levels on growth performance, serum variables, and bone characteristics in 1–28-day-old geese. One-day-old male Jiangnan White Geese ($n$ = 360) were randomly divided into five groups. Five corn-soybean diets were used in these groups, with NPP levels at 0.18%, 0.28%, 0.39%, 0.47%, 0.59% in the feed, respectively. The average body weight, serum phosphorus (P) content, tibia strength, tibia P content, and femoral skim weight were significantly reduced, by 0.18%, compared with 0.28–0.59%. These variables had significant linear fit ($p < 0.001$) between the levels of 0.18% and 0.58%. The estimated requirement of NPP for goslings is between 0.28% and 0.35%, based on the broken-line model, using the above indicators. In conclusion, the dietary NPP requirement for 1–28-day-old geese should be no less than 0.35% to ensure normal growth performance and bone development, based on body weight gain, serum P content, and skeletal variables.

**Keywords:** goslings; non-phytate phosphorus requirement; growth performance; bone mineralization; serum

## 1. Introduction

Phosphorus (P) is one of the essential mineral elements naturally present in many foods and also as supplements for the cellular activities, structural composition and metabolic regulation of organisms [1]. However, a considerable amount of P is present in plant-based diets in phytate P [2]. Phytate P is poorly available to poultry due to a lack of, or insufficient amount, of endogenous phytase to degrade phytate effectively [3,4]. Therefore, it requires inorganic P or phytase supplementation in the feed to support the health and performance of animals [3,4]. Insufficient inorganic P in diets may lead to hypophosphatemia [5]. It may also affect bone mineralization, causing skeletal cartilage dysplasia and rickets [6,7]. During the rearing of geese, it is necessary to ensure that they receive an adequate P intake and, thus, maximize their performance. Excessive addition of non-phytate phosphorus (NPP) aggravates the consumption of phosphate and increases pollution in the environment [6,8]. Therefore, it is essential to set a reasonable NPP requirement for geese, to save costs on feed and protect the environment. In recent years, breeding techniques have significantly improved the growth potential, calling for matching NPP requirements in geese. However, few recommendations on the NPP requirements of geese could be found. The available P in feed is recommended to be 0.42% on day 1 to day 28 and 0.37% on day 29 to day 70 in the Yangzhou Goose feeding Standard [9]. In practice, the dietary NPP level is

usually set according to the feeding standards of chickens in China, due to a lack of well-established standards for geese NPP requirement. Therefore, it is necessary to study the NPP requirements of geese to provide data to support the geese production industry.

About 85% of the P in the organisms of animals is present in the bones [6]. Diets with inadequate P levels can lead to skeletal abnormalities and negatively affect growth performance [10,11]. As an essential and critical mineral in poultry's diets, P performs critical functions in bone mineralization. Therefore, bone characteristics are commonly used to evaluate the mineralization of bones in chickens [12]. Body weight and skeletal characteristics are critical for obtaining NPP requirements for broilers [13]. It is worth noting that bone mineralization requires higher dietary NPP levels than growth [13]. The dietary NPP level was the highest to meet the need for maximum tibia ash deposition [14]. A correlation was observed between dietary NPP levels and tibial strength, tibial calcium and P content, femoral ash content, femoral P content, and toe P content [13–17]. Dietary NPP requirements for optimal body weight gain (BWG), feed conversion ratio (FCR), and bone mineralization are estimated to be 0.45% in broiler chickens from 1 to 21 days [17]. The predicted NPP requirements for BWG, serum P content, and tibia ash were 0.44%, 0.45%, and 0.41% of feed, respectively, in broilers from 3 to 30 days [18]. In another study, the predicted NPP requirements for tibia ash, BWG, and FCR were 0.33%, 0.19%, and 0.16% of feed, respectively, in broilers from 3 to 6 weeks [19]. In general, the NPP requirements for different poultry species range from 0.15% to 0.50% [13–19]. Thus, we assumed that the NPP requirement for geese was between 0.18% and 0.58%. Jiangnan White goose is a medium-sized goose line developed by our team. The common rearing period of medium-sized goslings is divided into two stages in China. The first stage is from 1 to 28 days, and the second stage is from 29 days to the listing age. In addition, days 1–28 are an important stage for the skeletal and weight development of the goslings [20,21]. Thus, we aimed to evaluate the dietary NPP requirement in 1 to 28-day-old geese, based on growth performance, serum variables, and bone characteristics.

## 2. Materials and Methods

### 2.1. Animal and Housing

The procedures in this paper were approved by the Institutional Animal Care and Use Committee of Yangzhou University (ethical protocol code: YZUDWSY 202103008) and conducted according to the relevant animal welfare regulations.

This experiment was conducted at the experimental farm of Yangzhou University (Yangzhou, China) from May to June 2021. A total of 360 healthy male Jiangnan White geese at one day old with similar body weight (BW) from the same flock were obtained from a commercial hatchery (Changzhou Four Seasons Poultry Industry Co., Ltd., Jintan, China). The birds were randomized to 5 dietary treatments, including 6 replicate pens and 10 geese per pen. All geese were reared in plastic wire-floor pens (2.1 m × 1.2 m). A three-liter waterer was provided to each pen during days 1–10, and freshwater was changed once a day. Water was supplied to goslings ad libitum via a nipple-type waterer on days 10–28. Mash feed was provided ad libitum throughout the trial. The temperature for the first four weeks was 29–31 °C, 27–29 °C, 24–26 °C, and 22–25 °C, respectively. Goslings were exposed to light for 23 h per day in the first week and 18 h per day in weeks 2 to 4.

### 2.2. Diets

A basal corn-soybean meal diet was formulated to provide adequate nutrients (except P) for all the groups. The nutritional level of the diet was based on a gosling's nutrient requirements from National Research Council (NRC, 1994) [22] and our laboratory's achievements over the years [20,21,23]. The basal diet without inorganic P was designed with an NPP level of 0.06%. The treatment diets were made by adding 0.50%, 0.94%, 1.38%, 1.81%, 2.24% calcium dihydrogen phosphate ($Ca(H_2PO_4)_2 \bullet H_2O$) in the basal diet, respectively with calculated NPP levels of 0.18%, 0.28%, 0.38%, 0.48%, 0.58% in the diets. Accordingly, the measured NPP levels in these treatments were 0.18%, 0.28%, 0.39%,

0.47% and 0.59%, respectively, in the diets. Calcium (Ca) and the weight of the feeds were balanced by limestone and vermiculite, respectively. Mash feed was used in the experiment. Diet and nutrient composition levels are shown in Table 1. The feed-grade $Ca(H_2PO_4)_2 \bullet H_2O$ was bought from Guzhou Chuanhen Chemical Corporation (Gui-zhou, China). The limestone, $Ca(H_2PO_4)_2 \bullet H_2O$, vermiculite, NaCl, methionine, lysine, and pre-mix were evenly mixed to form a mixture. This was then mixed with other feed ingredients to make the compound feed.

**Table 1.** Composition and nutrient levels of experimental diets (air-dry basis).

| Items | Dietary Non-phytate Phosphorus (NPP) Level | | | | |
|---|---|---|---|---|---|
| | 0.18% | 0.28% | 0.37% | 0.47% | 0.59% |
| Ingredients, % | | | | | |
| Corn, % | 58.30 | 58.30 | 58.30 | 58.30 | 58.30 |
| Soybean meal (CP [2], 43%), % | 31.60 | 31.60 | 31.60 | 31.60 | 31.60 |
| Wheat bran, % | 2.60 | 2.60 | 2.60 | 2.60 | 2.60 |
| Rice husk, % | 2.00 | 2.00 | 2.00 | 2.00 | 2.00 |
| Lime stone, % | 1.75 | 1.58 | 1.42 | 1.26 | 1.10 |
| Calcium dihydrogen phosphate, % | 0.50 | 0.94 | 1.38 | 1.81 | 2.24 |
| Vermiculite, % | 1.75 | 1.48 | 1.20 | 0.93 | 0.66 |
| DL–methionine, % | 0.20 | 0.20 | 0.20 | 0.20 | 0.20 |
| NaCl, % | 0.30 | 0.30 | 0.30 | 0.30 | 0.30 |
| Premix [1], % | 1.00 | 1.00 | 1.00 | 1.00 | 1.00 |
| Total, % | 100.0 | 100.0 | 100.0 | 100.0 | 100.0 |
| Nutrient levels [3] | | | | | |
| ME (MJ/kg) | 11.24 | 11.24 | 11.24 | 11.24 | 11.24 |
| CP, % | 18.29 | 18.37 | 18.44 | 18.40 | 18.38 |
| Crude fiber, % | 4.27 | 4.27 | 4.27 | 4.27 | 4.27 |
| Calcium (Ca), % | 0.78 | 0.75 | 0.77 | 0.78 | 0.76 |
| Total phosphorus (TP), % | 0.46 | 0.56 | 0.67 | 0.75 | 0.87 |
| Non-phytate P (NPP), % | 0.18 | 0.28 | 0.39 | 0.47 | 0.59 |
| Methionine, % | 0.49 | 0.49 | 0.49 | 0.49 | 0.49 |
| Lysine, % | 1.00 | 1.00 | 1.00 | 1.00 | 1.00 |

[1] Per kg of the premix contained the following items: vitamin A, 900,000 IU; vitamin D, 300,000 IU; vitamin E, 1800 IU; vitamin K, 150 mg; vitamin B1, 90 mg; vitamin B2, 800 mg; vitamin B6, 320 mg; vitamin B12, 1.2 mg; nicotinic acid, 4.5 g; pantothenic acid, 1100 mg; folic acid, 65 mg; biotin, 5 mg; Mn (manganese sulfate), 9.5 g; Zn (sodium selenite), 9 g; Fe (ferrous sulfate), 6 g; Cu (copper sulfate), 1 g; I (potassium iodide), 50 mg; Se (sodium selenite), 30 mg. [2] CP, crude protein; ME, metabolic energy. [3] Data of CP, Ca, TP, and NPP were analyzed, and other data were calculated.

### 2.3. Growth Performance and Sample Collection

All goslings were weighed at 8 a.m. at 14 and 28 days of age, and feed intake was recorded to calculate ADG (average daily gain), ADFI (average daily feed), and feed/gain (F/G) [23]. Then, one goose from each pen with BW similar to the mean weight of the pen ($\pm 100$ g) was selected and then weighed. Blood was collected via vacuum tubes from the leg veins of the geese and kept at room temperature for 30 min. The serum was produced by centrifuging the blood at 3500 r/min for 10 min with a frozen centrifuge (Cence DL-5M, Hunan Xiangyi Laboratory Instrument Development Co., Ltd., Hunan, China). Serum Ca, serum P, and serum alkaline phosphatase (ALP) were measured using a Synchron Fully Automated Biochemistry Analyzer (Hitachi 7180, Hitachi., Ltd., Tokyo, Japan). The geese were sacrificed by cutting the neck after electrical stunning at 86 mA for 18 s. The tibias, femurs and the third phalanx were collected from the left part of the body and refrigerated at $-20\ ^\circ$C until further analysis.

### 2.4. Skeleta Weight, Length, Strength, and Specific Gravity

Frozen tibias, femurs, and third phalanx (the middle toe bone) were thawed by leaving them on plastic plates at room temperature for one hour. The tibia bone and femur bone were defleshed, the patella with the toe cuticles were removed. The weight and length of the tibias and femurs were measured with a scale and a vernier caliper. Tibia bone

strength was measured with a dual column universal testing system (Instron 3367, Instron Corporation, Norwood, MA, USA). The determination methods were conducted according to Kääntee [24]. The calculation formula was N = W/V, where N is the specific gravity (g/cm$^3$), W is the tibial skim weight (g), and V is the bone volume (cm$^3$).

### 2.5. Skeletal Ash, Ca, and P Content

After determination of the fresh weight of bones (tibia, femur, and third phalanx), the bone was dried at 65 °C for 24 h, and defatted with ether for 7 d. The defatted bone was dried in an oven at 65 °C for 24 h and weighed to determine the skim weight. The dried bones were crushed (exact to 0.0001 g), charred in an electric furnace, and were burned in a muffle furnace at 550–580 °C for 12 h (SX2-4-10, Shanghai Gengfa Pharmaceutical Equipment Co., Ltd., Shanghai, China). The ash content of the bone samples was weighed and expressed as a percentage of skim bone weight. Skeletal ash was determined per GB/T6438–2002 [25]. The Ca was determined by the ethylenediaminetetraacetic acid method based on the GB/T6436–2018 [26]. The P was determined via the molybdenum yellow colorimetry method as per the GB/T6437–2018 [27]. Phytate P was determined by the ferric precipitation method [28,29].

### 2.6. Measurement of Feed Ingredients

A total of 2.0 kg of feed samples were collected from 10 positions after feed processing. Then 100.0 g feed was obtained from the 2.0 kg feed sample by the Quartering method. The feed sample was ground to pass through a 40-mesh sieve, mixed, dried at 65 °C for 2 h, cooled in the air to room temperature, and then stored in a closed container for further analysis. Crude protein was measured by the Kjeldahl method with the Kjeltec System 8400 instrument (SBS800, FOSS NIR Systems., Inc., Hillerød, Denmark). The feed sample was charred in an electric furnace and was burned in a muffle furnace at 550–580 °C for four hours. Ash was determined according to the GB/T6438–2002 [25]. The Ca was determined by the ethylenediaminetetraacetic acid method based on the GB/T6436–2018 [26]. The P was determined via the molybdenum yellow colorimetry method as per the GB/T6437–2018 [27]. Phytate P was determined by the ferric precipitation method [28,29].

### 2.7. Statistical Analysis

Raw data were organized using Microsoft Excel 2016 of Microsoft Corp. (Beijing, China). Data were analyzed using one-way ANOVA via SPSS (ver. 20.0) for Windows from SPSS, Inc. (Chicago, IL, USA). The data analysis results are expressed as the mean value and pooled standard error of the mean (SEM). The difference was significant if $p \leq 0.05$ by Duncan's multiple range tests. The broken-line models were set up by conducting nonlinear regression analyses using the PROC NLIN procedure of SAS 9.4 (SAS Institute Inc., Cary, NC, USA). The breaking points of these broken-line models were estimated and defined as dietary NPP requirements of gosling. The broken-line model used the equation Y = L + U (R − X), where the L is the corresponding Y value at the inflection point, U is the slope, and R is the inflection point. The R of these curves was calculated and expressed as the requirement of dietary NPP level of gosling. Some data were analyzed using quadratic regression analysis. The highest points of these curves were calculated and described as the optimal dietary NPP level of gosling.

## 3. Results

### 3.1. Growth Performance

Dietary NPP levels affected ($p < 0.05$) the BW of 28-day-old goslings (Table 2). The BW at day 28 was reduced ($p < 0.05$) at a dietary NPP level of 0.18%, with no difference ($p > 0.05$) among other levels between 0.28% and 0.59%. The F/G was not different ($p > 0.05$) among groups on days 14 and 28. The BW, ADG, and ADFI on days 14 and 28 were unaffected when NPP was between 0.28% and 0.59% ($p > 0.05$).

**Table 2.** Effects of dietary non-phytate phosphorus (NPP) levels on the growth performance of gosling at 14 and 28 days of age.

| Items | Dietary NPP Level | | | | | SEM | p Values | | |
|---|---|---|---|---|---|---|---|---|---|
| | 0.18% | 0.28% | 0.39% | 0.47% | 0.59% | | NPP Level | Linear | Quadratic |
| BW, g | | | | | | | | | |
| Day 14 | 604.7 | 654.8 | 627.3 | 656.9 | 666.1 | 8.06 | 0.082 | 0.025 | 0.693 |
| Day 28 | 1695 [b] | 1812 [a] | 1802 [a] | 1818 [a] | 1824 [a] | 12.19 | 0.001 | <0.001 | 0.016 |
| ADG, g/bird·d | | | | | | | | | |
| Day 14 | 33.51 | 37.59 | 36.13 | 37.08 | 38.06 | 0.62 | 0.147 | 0.050 | 0.448 |
| Day 28 | 55.24 | 59.43 | 57.69 | 58.25 | 58.80 | 0.66 | 0.241 | 0.122 | 0.489 |
| ADFI, g/bird·d | | | | | | | | | |
| Day 14 | 71.25 | 73.39 | 73.02 | 75.97 | 76.33 | 1.18 | 0.652 | 0.149 | 0.982 |
| Day 28 | 112.5 | 118.4 | 117.8 | 118.3 | 119.3 | 2.07 | 0.897 | 0.515 | 0.513 |
| F/G, g/g | | | | | | | | | |
| Day 14 | 2.03 | 1.96 | 1.98 | 1.93 | 1.93 | 0.02 | 0.536 | 0.124 | 0.725 |
| Day 28 | 2.04 | 2.00 | 2.04 | 2.03 | 2.03 | 0.03 | 0.992 | 0.894 | 0.862 |

[a,b] Values within a row without common superscripts differ significantly ($p \leq 0.05$). BW, Body weight; ADG, average daily gain; ADFI, average daily feed gain; F/G, ADFI/ ADG.

### 3.2. Serum Ca, P Contents, and ALP Activity

Serum P content was reduced ($p < 0.05$) at days 14 and 28, whereas serum ALP activity was increased ($p < 0.05$) at day 28 in the group of 0.18% compared with other groups (Table 3). Serum P content was not different ($p > 0.05$) among groups with dietary NPP levels between 0.39% and 0.59%. The serum P content in goslings reached a plateau at NPP levels above 0.39% on both days 14 and 28. Furthermore, serum P content showed a linear relationship with NPP levels between 0.18% and 0.59% in the diet ($p < 0.001$). Serum ALP activity was not different ($p > 0.05$) among groups with dietary NPP levels between 0.28% and 0.59%. The goslings fed with an NPP of 0.39% had the lowest serum ALP activity. Serum Ca content was not affected by NPP levels between 0.18% and 0.59% ($p > 0.05$).

**Table 3.** Effects of dietary non-phytate phosphorus (NPP) levels on serum variables of goslings at 14 and 28 days of age.

| Items [1] | Dietary NPP Level | | | | | SEM | p Values | | |
|---|---|---|---|---|---|---|---|---|---|
| | 0.18% | 0.28% | 0.39% | 0.47% | 0.59% | | NPP Level | Linear | Quadratic |
| Serum Ca content, mmol/L | | | | | | | | | |
| Day 14 | 2.78 | 2.61 | 2.48 | 2.49 | 2.53 | 0.040 | 0.105 | 0.013 | 0.244 |
| Day 28 | 2.52 | 2.32 | 2.35 | 2.36 | 2.42 | 0.026 | 0.074 | 0.057 | 0.097 |
| Serum P content, mmol/L | | | | | | | | | |
| Day 14 | 1.98 [b] | 2.34 [a] | 2.59 [a] | 2.53 [a] | 2.58 [a] | 0.064 | 0.003 | 0.001 | 0.038 |
| Day 28 | 1.46 [c] | 1.71 [b] | 1.87 [a] | 1.89 [a] | 1.87 [a] | 0.035 | <0.001 | <0.001 | 0.001 |
| Serum ALP activity, U/L | | | | | | | | | |
| Day 14 | 883.8 | 796.2 | 741.7 | 815.0 | 744.0 | 22.82 | 0.266 | 0.110 | 0.395 |
| Day 28 | 1120 [b] | 908.0 [a] | 803.8 [a] | 819.8 [a] | 855.8 [a] | 28.49 | <0.001 | <0.001 | 0.001 |

[a–c] Values within a row without common superscripts differ significantly ($p \leq 0.05$). [1] Ca: calcium; P, phosphorus; ALP, alkaline phosphatase.

### 3.3. Bone Characteristics

Dietary NPP levels affected ($p < 0.05$) the tibia variables in 28-day-old goslings (Table 4). The tibia's length, strength, ash content, and P content were reduced with an NPP level of 0.18% compared with other groups ($p < 0.05$). Dietary NPP levels of 0.18% reduced ($p < 0.05$) the skim weight and specific gravity of the tibia compared with other groups with

levels between 0.39% and 0.59%. No difference was observed in the tibia's skim weight and the specific gravity between 0.18% and 0.28% ($p > 0.05$). The above indicators showed a linear relationship with NPP levels between 0.18% and 0.59% ($p < 0.05$). No difference was observed in any bone variables among groups of 0.39%, 0.47%, and 0.59% ($p > 0.05$).

**Table 4.** Effects of dietary non-phytate phosphorus (NPP) levels on the tibia variables in 28-day-old goslings.

| Items [1] | Dietary NPP Level | | | | | SEM | p Values | | |
|---|---|---|---|---|---|---|---|---|---|
| | 0.18% | 0.28% | 0.39% | 0.47% | 0.59% | | NPP Level | Linear | Quadratic |
| Length, cm | 11.06 [b] | 11.58 [a] | 11.61 [a] | 11.72 [a] | 11.76 [a] | 0.08 | 0.016 | 0.003 | 0.122 |
| Width, cm | 0.795 | 0.798 | 0.818 | 0.803 | 0.826 | 0.06 | 0.483 | 0.116 | 0.447 |
| Fresh weight, g | 10.21 | 10.27 | 10.62 | 11.00 | 11.19 | 0.15 | 0.160 | 0.014 | 0.814 |
| Skim weight, g | 4.98 [c] | 5.36 [bc] | 5.70 [ab] | 5.71 [ab] | 6.06 [a] | 0.10 | 0.004 | <0.001 | 0.576 |
| Bone strength, N | 267 [b] | 334 [a] | 376 [a] | 385 [a] | 386 [a] | 12.18 | 0.002 | <0.001 | 0.045 |
| Volume, mL | 8.00 | 7.78 | 7.88 | 8.17 | 7.83 | 0.13 | 0.916 | 0.961 | 0.967 |
| Specific gravity, g/cm³ | 1.28 [b] | 1.32 [ab] | 1.34 [a] | 1.35 [a] | 1.35 [a] | 0.01 | 0.020 | 0.002 | 0.185 |
| Ash, % | 49.93 [b] | 52.50 [a] | 53.30 [a] | 53.48 [a] | 53.72 [a] | 0.37 | 0.006 | 0.001 | 0.043 |
| Ca, % | 16.96 | 18.28 | 18.61 | 18.39 | 18.52 | 0.29 | 0.367 | 0.123 | 0.230 |
| P, % | 7.96 [b] | 8.65 [a] | 8.99 [a] | 8.96 [a] | 9.00 [a] | 0.10 | 0.001 | <0.001 | 0.013 |

[a-c] Values within a row without common superscripts differ significantly ($p \leq 0.05$). [1] Ca: calcium; P: phosphorus.

The femur's length, fresh weight, and skim weight were reduced ($p < 0.05$), with a dietary NPP level of 0.18% as compared with other groups (Table 5). No difference was observed in the femur variables in the other groups with NPP levels between 0.28% and 0.59% ($p > 0.05$). The length, fresh weight, skim weight of the femur showed a linear relationship with dietary NPP levels between 0.18% and 0.59% ($p < 0.05$). The ash content of the third phalanx was reduced ($p < 0.05$) with a dietary NPP level of 0.18% as compared with other NPP levels (Figure 1). The Skim weight of the third phalanx was reduced ($p < 0.05$) with a dietary NPP level of 0.18% compared with the three dietary NPP levels between 0.39% and 0.59%. No difference was observed in the skim weight between 0.18% and 0.28% ($p > 0.05$). The variables of the third phalanx did not differ with NPP levels among 0.39% and 0.59% ($p > 0.05$). The third phalanx's skim weight and ash content showed a linear relationship with dietary NPP levels between 0.18% and 0.59% ($p < 0.05$).

**Table 5.** Effects of dietary non-phytate phosphorus (NPP) levels on the femur variables of 28-day-old goslings.

| Items [1] | Dietary NPP Level | | | | | SEM | p Values | | |
|---|---|---|---|---|---|---|---|---|---|
| | 0.18% | 0.28% | 0.39% | 0.47% | 0.59% | | NPP Level | Linear | Quadratic |
| Length, cm | 6.07 [b] | 6.43 [a] | 6.48 [a] | 6.56 [a] | 6.56 [a] | 0.04 | <0.001 | <0.001 | 0.007 |
| Width, cm | 0.81 | 0.80 | 0.82 | 0.81 | 0.83 | 0.01 | 0.521 | 0.228 | 0.342 |
| Fresh weight, g | 4.96 [b] | 5.78 [a] | 6.04 [a] | 6.09 [a] | 6.07 [a] | 0.12 | 0.003 | 0.001 | 0.022 |
| Skim weight, g | 2.34 [b] | 2.89 [a] | 3.07 [a] | 3.03 [a] | 3.07 [a] | 0.06 | <0.001 | <0.001 | 0.001 |
| Ash, % | 47.56 | 50.21 | 50.32 | 50.57 | 50.62 | 0.45 | 0.156 | 0.043 | 0.171 |
| Ca, % | 18.83 | 19.83 | 20.73 | 20.34 | 20.23 | 0.24 | 0.096 | 0.041 | 0.066 |
| P, % | 8.42 | 9.18 | 9.37 | 9.42 | 9.65 | 0.15 | 0.073 | 0.009 | 0.298 |
| Ash Ca, % | 38.39 | 39.50 | 40.63 | 40.24 | 40.27 | 0.38 | 0.799 | 0.776 | 0.242 |
| Ash P, % | 17.77 | 18.42 | 18.67 | 18.62 | 18.54 | 0.24 | 0.379 | 0.107 | 0.258 |

[a,b] Values within a row without common superscripts differ significantly ($p \leq 0.05$). [1] Ca, calcium; P, phosphorus.

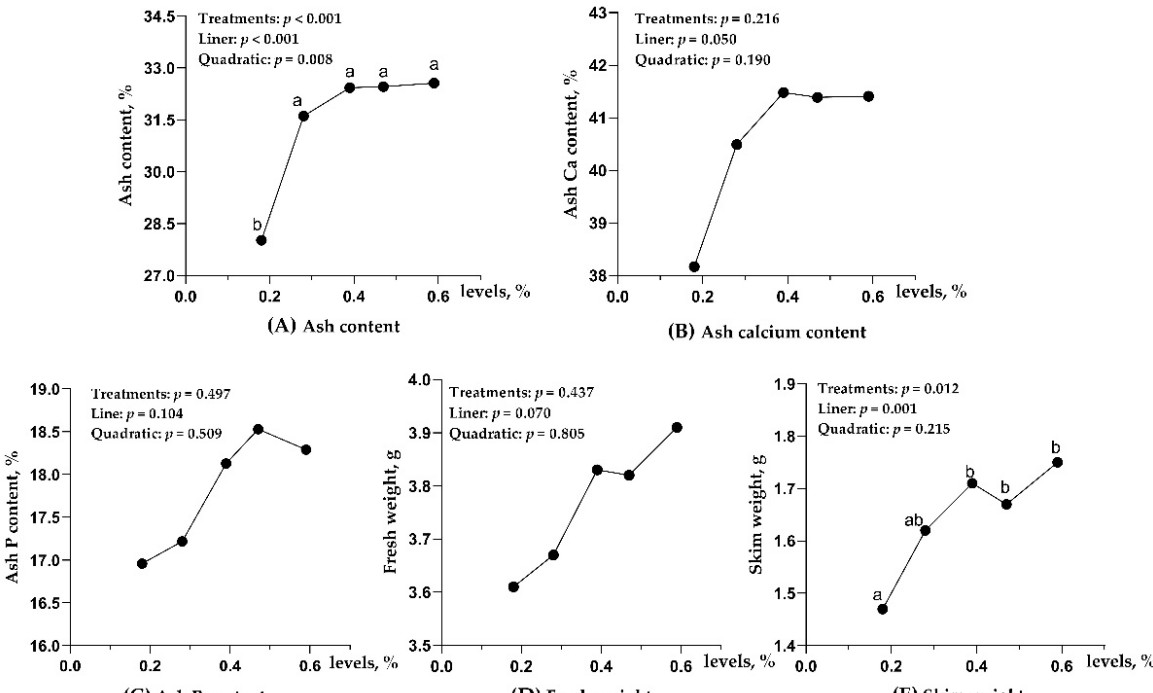

**Figure 1.** Effects of dietary non-phytate phosphorus (NPP) levels on the third phalanx variables in 28-day-old goslings. [a,b] Values within a row without common superscripts differ significantly ($p \leq 0.05$). P: phosphorus.

### 3.4. The NPP Requirement of Goslings

The NPP requirement for the 1 to 28-day-old goslings was obtained by fitting the relevant data, using the broken-line model, with an equation of $Y = L + U (R - X)$. The L is the corresponding Y value at the inflection point, U is the slope, and R is the inflection point (Table 6). The NPP requirement for moderate BW, serum P, tibia strength, tibia P content, tibia ash content, femur fresh weight, and skim weight of the third phalanx was 0.28%, 0.35%, 0.35%, 0.33%, 0.33%, 0.31%, 0.31%, respectively, based on the broken-line model ($p \leq 0.05$). The NPP requirement for optimal BW, serum P content, tibia bone strength, tibia ash content, tibia P content, and the third phalanx's ash was 0.48%, 0.49%, 0.51%, 0.50%, 0.49%, 0.46%, respectively, based on the quadratic regressions ($p \leq 0.05$, Table 6).

**Table 6.** Dietary non-phytate phosphorus (NPP) requirement for 1- to 28-day-old goslings based on two regression models.

| Items | Regression Equations | $p$ Values | $R^2$ | NPP Requirements, % |
|---|---|---|---|---|
| Broken-line regression | | | | |
| Body weight (BW), g | $Y = 1814.8 - 1162.5 (0.2829 - x)$ | <0.001 | 0.52 | 0.28 |
| Serum phosphorus (P), % | $Y = 1.8759 - 2.5167 (0.3453 - x)$ | <0.001 | 0.77 | 0.35 |
| Tibia bone strength, N | $Y = 382.3 - 670.8 (0.3524 - x)$ | <0.001 | 0.48 | 0.35 |
| Tibia P, % | $Y = 8.9819 - 6.9333 (0.3267 - x)$ | <0.001 | 0.53 | 0.33 |
| Tibia ash, % | $Y = 53.4994 - 25.6998 (0.3267 - x)$ | 0.010 | 0.40 | 0.33 |
| Femur fresh weight, g | $Y = 3.0576 - 5.45 (0.3117 - x)$ | <0.001 | 0.67 | 0.31 |
| Third phalanx's skim weight, g | $Y = 1.6633 - 2.1267 (0.3029 - x)$ | <0.001 | 0.72 | 0.30 |
| Quadratic regression | | | | |
| BW, g | $Y = 1527.2 + 1250.7 x - 1290.8 x^2$ | <0.001 | 0.45 | 0.48 |
| Serum P, % | $Y = 0.815 + 4.42 x - 4.471 x^2$ | <0.001 | 0.70 | 0.49 |

**Table 6.** *Cont.*

| Items | Regression Equations | *p* Values | R$^2$ | NPP Requirements, % |
|---|---|---|---|---|
| Tibia bone strength, N | Y = 98.033 + 1146.4 x − 1121.1 x$^2$ | <0.001 | 0.44 | 0.51 |
| Tibia ash, % | Y = 44.91 + 35.35 x − 35.01 x$^2$ | 0.001 | 0.34 | 0.50 |
| Tibia P, % | Y = 6.427 + 10.755 x − 10.974 x$^2$ | <0.001 | 0.48 | 0.49 |
| Third phalanx's ash, % | Y = 21.015 + 51.993 x − 55.929 x$^2$ | 0.001 | 0.38 | 0.46 |

## 4. Discussion

### 4.1. Growth Performance

Phosphorus (P) is essential to sustaining life, largely through the growth and development of goslings. It is a critical component of nucleic acids and phospholipids, significantly affecting growth and development [30]. In the present study, the BW was significantly reduced by a dietary NPP level of 0.18% compared with other levels (0.28–0.59%). However, the growth performance was not further affected when the NPP level was above 0.28%. Similarly, feed intake and BW were not significantly affected by NPP levels above 0.30% (0.30%, 0.5% and 0.7%) in geese [31]. The continuous increase in dietary P level has no significant effect on the production performance of broilers in the diet, with a total P level of 0.42% [14]. Feed intake and BWG decreased by 25% and 33% in broilers, respectively, when the dietary available P level decreased from 0.33% to 0.12% [32]. A dietary NPP level of 0.06% significantly reduced broilers' BWG and FCR by 68.7% and 24%, respectively, compared with 0.44% [33]. The ADG and feed intake of broilers were significantly lower with a dietary NPP level of 0.25% than those with 0.35%, 0.40%, and 0.45% [18].

Similarly, the dietary NPP level of 0.15% significantly reduced the BW of 10- to 22-day-old broilers, based on a consistent Ca level in the diets [34]. Consistent with our results, no beneficial effect on growth performance was observed in broiler chicks when the NPP levels were between 0.45% and 0.60% [35]. It is clear from this study that inadequate NPP in the feed could hamper the growth of geese; thus, a reduction in feed intake, resulting in the birds' impaired growth and development [10,11]. Another possible reason for the reduced growth performance of goslings could be due to the hampered skeletal development (tibia and femur). Similar to our findings, Ayres et al. [36] reported that shorter tibia and femur decreased the growth performance of broilers. However, dietary NPP levels did not significantly affect the F/G of goslings in our study. One possible reason might be that both body weight and feed intake were reduced in goslings at a dietary NPP level of 0.18%, leading to no significant difference in F/G. Similar phenomena were observed in chickens in [9]. Even if the F/G was not different among groups, a faster growth rate in treatments of 0.28–0.59% means an earlier time on the market and savings of management costs.

The dietary NPP level of 0.28% was sufficient to meet the requirement for growth from 1 to 28 days of age, based on the broken-line model in our study. This result is consistent with the result analyzed via the one-way ANOVA. However, 0.48% was the dietary NPP level to meet the optimum growth requirement of the goslings from 1 to 28 days of age, based on a quadratic function regression. The NPP requirement was suggested to be 0.34% for Arbor Acres broilers [13]. In another study, the NPP requirements were 0.44%, 0.45%, and 0.74% for broilers, respectively, by fitting the body weight, serum P content, and tibial ash content with the polynomial equation [18]. Bone mineralization had a higher demand for dietary P content than weight gain [14]. The NPP requirement of 0.28% was similar but slightly lower than the NRC (1994) recommendation (0.30%) in geese [20]. The best growth performance was achieved at a dietary NPP level of 0.60%, even better than those in the 0.90% group [37]. Possible reasons for the difference among studies might be due to different animal breeds, species, and judging variables. The optimum NPP level was 0.48% in our study, based on a quadratic function regression analysis, higher than the recommendation of NRC (1994) [20] and Liu [13]. However, the difference of BW between the groups of 0.28% and 0.47% was only 6.0 g (0.30%) in the present study. In addition,

a higher dietary NPP level causes more P excretion and pollution in the environment and costs more money [14,19]. Thus, the requirement of dietary NPP level for growth performance is suggested to be 0.28% in 1- to 28-day-old geese.

### 4.2. Serum P Contents and ALP Activity

The ALP is a group of isoenzymes, widely distributed in the human liver, bones, and intestines. The ALP promotes bone mineralization and is a marker of bone formation. In general, serum ALP levels are maintained in a stable range in animals. The bone ALP levels could be altered when the balance between the bone resorption and formation in the skeleton is affected, leading to the change in serum ALP levels [13,15]. The serum ALP activity was increased in the group of 0.18% compared to other groups with dietary NPP levels between 0.28% and 0.59%. Goslings in the 0.30% group had significantly higher serum ALP activity than those in the 0.50% and 0.70% groups [38]. Similarly, Du et al. observed that the ALP activity was reduced linearly to increased dietary NPP levels [37]. The serum ALP activity was also increased with low dietary NPP levels in broilers [13]. Broilers consuming low Ca- and P-level feeds have significantly higher serum ALP activity than normal controls [39].

The P absorption in the intestine and P excretion in the kidneys work together to maintain a relatively stable P level in the body's internal environment [5]. Thus, the serum P content is relatively stable under normal conditions. The serum P content of geese was reduced considerably with a low dietary NPP level of 0.18%. Inadequate dietary P may lead to poor P retention, affecting serum P content [18]. Enough NPP should be contained in the diet to ensure a relatively stable serum P level and maintain the homeostasis of the internal environment [40]. The goslings fed with an NPP at both low level (0.30%) and high level (0.70%) in feeds had lower serum P levels than those who had a moderate level (0.50%) of NPP in feed [38]. When the dietary NPP level was lower than 0.33%, the serum P content of broilers was significantly lower than the groups with higher P levels among 0.38–0.58% [13]. Consistently, the serum P was reduced by an NPP level of 0.23% compared with 0.45% in 21-day-old broilers [41]. Our data indicated that inadequate dietary NPP reduced the serum P content by decreasing the goslings' feed intake, leading to insufficient P intake and less circulating P. Another reason might be that the bones were in a rapid growth phase in goslings during 1 to 28 days of age. The P retention was more robust than the P release in the bones. To ensure a regular serum P content in goslings, we recommend that the NPP level in the diet should be at least 0.35%, judging from the serum P content by fitting with a broken-line model.

### 4.3. Bone Characteristics

Tibia quality is a critical indicator of bone development. Bone is the primary storage organ for P, accounting for about 85% of the body's total P [6]. Tibia ash content, tibia Ca, P content, and other tibia bone indicators were significantly correlated with dietary P levels [42]. Our results showed that the tibia's length, strength, P content, and ash content were significantly reduced with a dietary NPP level of 0.18% compared with levels above 0.28%. The goslings had significantly lower tibia ash, calcium, and P content with 0.30% NPP than those with 0.50% and 0.70% NPP in feeds [38]. Broilers' diet with 0.11%, 0.19%, and 0.27% of NPP resulted in lower tibial bone ash, tibial P, and Ca content than birds that had the higher NPP levels (0.35%, 0.43%, 0.51%, and 0.59%) [43]. Similarly, a dietary NPP level of 0.23% caused lower tibial ash content in broilers than the level of 0.45% [41]. However, the continued increase in dietary NPP content did not benefit tibial bone development [43]. Although dietary NPP levels greatly affected tibia strength [13], tibia ash, and P content [13,18], these variables did not differ when the NPP level was above 0.28%. One possible reason might be that the discharge of P through feces increased sharply after the diets met the need for optimum tibia ash [14,19]. The absorption of intestinal P may be regulated by the protein expression of type IIb sodium-dependent phosphate co-transporter (NaPi–IIb), inorganic phosphate transporter 1 (PiT-1), and inorganic phosphate

transporter 2 (PiT-2) [33]. In the present study, the NPP requirements for tibia strength, tibia P content, and tibia ash content were 0.35%, 0.33%, and 0.33%, respectively, analyzed by a broken-line model. These results were higher than the NPP requirement of 0.27% for BW. Consistently, the requirement for growth is lower than the requirement for tibia variables in chicken [13]. The dietary NPP level should be at least 0.35% to ensure a good tibia quality in goslings.

The femur connects to the tibia and the third phalanx, known as the third phalanx, and is one of the phalanges. The femur has a broader diameter than the tibia and metatarsus [44]. The development and mineralization of the femur is an essential factor affecting leg disease and growth performance [44]. Low dietary P levels significantly affect the femur and the third phalanx indicators, which are sensitive indicators to evaluate the NPP requirements in poultry [13,17,45–47]. Low P levels in the diet reduce femur quality and increase the incidence of femur fractures in broilers [46]. Consistent with our results, femur length, femur weight, and weight/length ratio were significantly lower in broilers in the lowest NPP group than in the other groups [48].

Inadequate dietary NPP levels significantly reduced the third phalanx skims weight and the third phalanx ash content in goslings. Consistent with our study, dietary NPP levels greatly affected broilers' third phalanx development and mineralization [13,17]. The toe bone ash content of broilers increased with increasing dietary NPP levels between 0.25% and 0.50%, in line with a linear and quadratic variation [17]. Dietary NPP levels significantly affected broilers' toe ash content and ash P content [13]. Reducing the NPP content of the diet had a significant adverse effect on the toe bone ash content in broiler chicks [45,47]. Broilers in the groups with lower dietary NPP levels had lower toe ash content and third phalanx ash P content than those with higher NPP levels [13]. In this trial, the NPP requirements of the goslings were 0.31% and 0.30%, respectively, fitting the broken-line model for the femur and the third phalanx skim weight. To ensure the healthy development of the femur and the third phalanx of goslings, we recommend that the NPP level of the diet should be at least 0.31%.

This experiment showed that the NPP level required to meet tibial growth was higher than the growth of the femoral and the body. The degree of bone mineralization is related to the growth rate of the broiler [49]. Changes in dietary NPP levels were not sufficient to significantly affect femur and third phalanx variables in the study of Waldroup et al. [14]. Similarly, femur and middle phalanx were not sensitively affected by NPP levels in our research. Opposingly, tibia variables were sensitively affected by NPP levels. One possible reason might be that the growth and mineralization of the femur and middle phalanx are slower than that of the tibia [14,44]. Healthy development of the femur and middle phalanx requires lower dietary NPP levels than the tibia.

*4.4. Overall Discussion*

The NPP requirements for goslings were 0.28%, 0.35%, 0.35%, 0.33%, 0.33%, 0.31% and 0.30%, respectively, based on the BW, serum P content, tibia strength, tibia ash content, tibia P content, femur fresh weight, and third phalanx skim weight. This range is higher than the NRC recommendation (0.30%) and similar to the recommendation (0.35%) reported by Alagawany et al. [50]. The optimal Ca/NPP range estimated was 2.03 to 2.23 (Ca: 0.78%, NPP: 0.35%, 0.39%), based on results of present study. Similarly, previous researchers' optimal Ca/NPP range varied from 2.0 to 2.17 [20,50,51]. The results obtained by fitting the quadratic curve model might indicate the optimum growth and bone performance. These results are higher than those from other scholars' studies [13,20,50,52]. Therefore, we estimated the minimum NPP requirement of 1- to 28-day-old geese via the broken-line model to get an economical and environmentally friendly feed formula.

**5. Conclusions**

In conclusion, the NPP requirement is between 0.28% and 0.35%, based on growth performance, serum variables, and bone characteristics. We recommend that the minimum

dietary NPP requirement for 1- to 28-day-old geese is 0.35%, to ensure stable growth performance, serum variables, and good skeletal quality.

**Author Contributions:** Data collection, animal trials, N.L. and G.S.; writing—original draft preparation, N.L. and L.X.; data analysis, N.L. and Y.C.; writing—review and editing, L.X. and Z.W.; funding acquisition, L.X. and H.Y. and Z.W. All authors have read and agreed to the published version of the manuscript.

**Funding:** This work was financially supported by the China Agriculture Research System of MOF and MARA, the Jiangsu Agriculture Science and Technology Innovation Fund (CX (18) 1004), and the Jiangsu Agriculture Industry Technology System of P. R. China (JATS (2021) 496).

**Institutional Review Board Statement:** The procedures in this paper were approved by the Institutional Animal Care and Use Committee of Yangzhou University (ethical protocol code: YZUDWSY 202103008) and conducted according to the relevant animal welfare regulations.

**Informed Consent Statement:** Not applicable.

**Data Availability Statement:** Data are available on reasonable request from the corresponding author.

**Conflicts of Interest:** The authors declare no conflict of interest.

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
