# Peer review of "Requirement of Non-Phytate Phosphorus in 1- to 28-Day-Old Geese Based on Growth Performance, Serum Variables, and Bone Characteristics"

_agriculture, doi:10.3390/agriculture12040479_

Round 1

Reviewer 1 Report

The introduction is insufficient, add more data about NPP requirements in chicken and other poultry

Line 47 separate the numerical value about the unit, check and unify in all manuscript

Line 73 indicate the NRC abbreviation

Add recent citations in sec 2.3

Check the model and origin of all devices

Detailed the methods in sec 2.6

Enhance the presentation of data, it should make a comparison between treatments, not mention the values in tables, check

Indicate the abbreviations in Table 2 and enhance their format

 I suggest merging Tables 7 and 8 into one table

Some data may convert to figures such as Table 6

The discussion is well handled, but the citations need to update to 2021-2022

Check the outputs of all references

Author Response

Comments (Com) 1: The introduction is insufficient, add more data about NPP requirements in chicken and other poultry.

Answer (Ans) 1: We added more data about NPP requirements in chicken and other poultry in lines 58-73 in the revised manuscript. Thank you very much for your suggestions!

Com 2: Line 47 separate the numerical value about the unit, check and unify in all manuscript

Ans 2: The numerical value about the unit was separated in the manuscript. Please see lines 76, 299, 321 in the revised manuscript.

Com 3: Line 73 indicate the NRC abbreviation.

Ans 3: Full name of the NRC was added in line 108.
Com 4:
Add recent citations in sec 2.3.

Ans 4: Some citation of the method was added in Sec 2.3. More details of measurement and calculation formula were also added. Please see lines 130-144 in the revised manuscript.
Com 5:
Check the model and origin of all devices.

Ans 5: All devices' models and origins were checked and supplemented. Please see lines 136, 140,152,162,174 in the revised manuscript.

Com 6: Detailed the methods in sec 2.6.

Ans 6: Details of the methods were added in sec 2.6. Please see lines 169-175 in the revised manuscript.

Com 7: Enhance the presentation of data. It should make a comparison between treatments, not mention the values in tables, check.

Ans 7: Some contents have been modified and added. Comparison among treatments could be found in lines207-215, 223-229, 236-249, 264-274.

Com 8: Indicate the abbreviations in Table 2 and enhance their format.

Ans 8: Abbreviations were indicated, and their format was enhanced in Table 2. Please see lines 204 and 205 in the revised manuscript.

Com 9: I suggest merging Tables 7 and 8 into one table.

Ans 9: Tables 7 and 8 were combined into Table 6.
Com 10: Some data may convert to figures such as Table 6.

Ans 10: Some data are represented in figures 1.

Com 11: The discussion is well handled, but the citations need to update to 2021-2022.

Ans 11: Thank you, your approval is what keeps us going. Some latest citations were added in the introduction and discussion, related references were highlighted in yellow in the reference section. Thank you very much for your suggestions!

Com 12: Check the outputs of all references.

Ans 12: All references have been carefully checked, and the changes were marked with highlights.

The major changes are highlighted. Track changes could find other minor changes or English revisions.

Reviewer 2 Report

Dear Authors

Regarding the manuscript title:  

 Requirement of Non–phytate Phosphorus in 1–to 28–day–old 2 Geese based on Growth Performance, Serum Variables, and 3 Bone Characteristics. The scientific background of the topic was well mentioned in the introduction part. The experiment design quite good, as well as the replicates and methods used, were quite good. The results obtained were presented in tables well discussed with other author’s results. However, there are some observation in the present paper should be corrected and add to improve the quality of the paper.

Introduction:

Need more information about Phosphorus sources in poultry diets.

Discussion

Need more explanation about effect of Non–phytate Phosphorus on growth performance.

Table 1. Composition and nutrient levels of experimental diets (air-dry basis)

Add the protein ratio of Soybean meal, %

Need more explanation why body weight was increased by high level but FCR not influenced?

Why Serum P content  was increased by high levels?

Author Response

Comments (Com) 1: Need more information about phosphorus sources in poultry diets.

Answer (Ans) 1: We added more information on phosphorus sources in poultry diets. Please see lines 33-37 in the revised manuscript. Thank you very much for your suggestions!

Com 2: Need more explanation about effect of Non–phytate Phosphorus on growth performance.

Ans 2: We added more information about the effect of Non–phytate Phosphorus on growth performance. Please see lines 308-312 in the revised manuscript.

Com 3: Table 1. Composition and nutrient levels of experimental diets (air-dry basis)

Add the protein ratio of Soybean meal, %

Ans 3: We added the protein ratio of Soybean meal in Table 1. Please see line 121 (Table 1) in the revised manuscript.

Com 4: Need more explanation why body weight was increased by high level but FCR not influenced?

Ans 4: We explained it in lines 312-314 in the revised manuscript.
Com 5: Why Serum P content was increased by high levels?

Ans 5: We added possible reasons why low NPP levels reduced serum P content in lines 368-372 in the revised manuscript.

The major changes are highlighted. Track changes could find other minor changes or English revisions.

Reviewer 3 Report

Abstract

Row 15. For that 360, 1-day-old

Introduction

Row 40, can you please mention what is the recommended level of NPP currently in China?

Material and Methods

How were the feed and water provided?

One important aspect is how the NPP was added. Was mixed directly into the feed, or was mixed into the premix and then the premix was added into the compounds feed.

Also, another aspect is that if the additional levels of NPP were added supplementary over the P already existed in the premix or how? Please add this information.

Row. 84. In the results in Table 2, the authors present BW at 14 and 28 days. But in the M&M chapter, they mention that the animals were weighed only at 28 days. I think that this aspect needs some clarification or just add the rest of the information. Further, it is worth mentioning how the BW, ADG, ADFI and F/G were calculated. If the authors followed the chickens I assume that these parameters were calculated as presented in the literature (i.e. https://doi.org/10.1080/1828051X.2020.1845576) or as other papers on gosling (i.e. https://doi.org/10.1016/j.psj.2021.01.037)  

Further, I think that the slaughtering explanation is missing in this paper. A paragraph about the slaughter technique must be added.

Row 86-91. How the serum was obtained? Firstly, you collected blood samples, which were centrifuged after to obtain the serum, so, please add this details.

Row 91 ... until further analyses.

Row 101-102 The process of drying was made 2 times? Please rephrase it; it is not very clear how the skim weight was measured.

Results

Row 15-159. In table 4 are presented the results of volume and specific gravity, but in the M&M chapter there is no word about how were determined. Please clarify.

Discussions

This chapter is well documented and the discussions reflect the obtained results.

General comment

It is not very clear if the 1 to 28 days is just a growing phase, like only the starter phase, or is starter with grower phase. Why did you decide to run a trial only for 28 days? This aspect is also missing in the introduction chapter. Please add a sentence explaining why is essential to provide an adequate level of P in the gosling diet in the first 1 to 28 days of raising.

Author Response

Comments (Com) 1: Row 15. For that 360, 1-day-old

Answer (Ans) 1: We have changed the words. Please see lines 15 and 16 in the revised manuscript.

Thank you very much for your suggestions!

Com 2: 14. Row 40, can you please mention what is the recommended level of NPP currently in China?

Ans 2: We added the recommended level of AP currently in China in lines 47 and 49 in the revised manuscript.

Com 3: How were the feed and water provided?

Ans 3: The method was added in lines 94-97 in the revised manuscript.
Com 4: One important aspect is how the NPP was added. Was mixed directly into the feed, or was mixed into the premix and then the premix was added into the compounds feed. 

Ans 4: The limestone, calcium dihydrogen phosphate, vermiculite, NaCl, methionine, lysine, and premix were evenly mixed to form a mixture. Then the mixture was mixed with other feed ingredients to make the compound feed. The information was added in lines 116-120. Thank you very much for your suggestions!

Com 5: Also, another aspect is that if the additional levels of NPP were added supplementary over the P already existed in the premix, or how? Please add this information.

Ans 5: The additional levels of NPP were added to a basal diet with an NPP of 0.06%, leading to the final NPP levels of 0.18%, 0.28%, 0.39%, 0.47% and 0.59% in the feeds (measured values). This information was added in lines 109-112

Com 6: Row. 84. In the results in Table 2, the authors present BW at 14 and 28 days. But in the M&M chapter, they mention that the animals were weighed only at 28 days. I think that this aspect needs some clarification or just add the rest of the information. Further, it is worth mentioning how the BW, ADG, ADFI and F/G were calculated. If the authors followed the chickens I assume that these parameters were calculated as presented in the literature (i.e. https://doi.org/10.1080/1828051X.2020.1845576) or as other papers on gosling (i.e. https://doi.org/10.1016/j.psj.2021.01.037) 

Ans 6: The literature of growth performance and calculation formula were added in line 132 in the revised manuscript.

Com 7: Further, I think that the slaughtering explanation is missing in this paper. A paragraph about the slaughter technique must be added

Ans 7: We added the slaughter technique of the information. Please see lines 141 and 142 in the revised manuscript.

Com 8: How the serum was obtained? Firstly, you collected blood samples, which were centrifuged after to obtain the serum, so, please add this details.

Ans 8: We added some information on how the serum was obtained in lines 134-137 in the revised manuscript.

Com 9: Row 91 ... until further analyses.

Ans 9: We added the "until further analyses" on line 144.
Com 10: Row 101-102 The process of drying was made 2 times? Please rephrase it; it is not very clear how the skim weight was measured.

Ans 10: The process of drying was made two times. We added some words to ensure that the sentences were better understood. Please see lines 158 and 159 in the revised manuscript.

Com 11: Row 15-159. In table 4 are presented the results of volume and specific gravity, but in the M&M chapter there is no word about how were determined. Please clarify.

Ans 11: We added the volume and specific gravity of the information in the M&M. Please see lines 152-155 in the revised manuscript.

Com 12: This chapter is well documented and the discussions reflect the obtained results.

General comment.

Ans 12: Thank you. Your approval gives us the motivation to work harder.

Com 13: It is not very clear if the 1 to 28 days is just a growing phase, like only the starter phase, or is starter with grower phase. Why did you decide to run a trial only for 28 days? This aspect is also missing in the introduction chapter. Please add a sentence explaining why is essential to provide an adequate level of P in the gosling diet in the first 1 to 28 days of raising.

Ans 13: We added the reasons for the feeding stage in lines 76-80 in the revised manuscript. Thank you very much for your suggestions!

The major changes are highlighted. Track changes could find other minor changes or English revisions.

Round 2

Reviewer 1 Report

The authors have carefully processed all comments. The quality of the manuscript has increased significantly. I have no further comments.